

# Identification of SPRR3 as a novel diagnostic/prognostic biomarker for oral squamous cell carcinoma via RNA sequencing and bioinformatic analyses

Lu Yu[1,*], Zongcheng Yang[1,*], Yingjiao Liu[2], Fen Liu[3], Wenjing Shang[3], Wei Shao[3], Yue Wang[3], Man Xu[4], Ya-nan Wang[1], Yue Fu[4] and Xin Xu[1]

[1] Department of Implantology, School and Hospital of Stomatology, Shandong University & Shandong Provincial Key Laboratory of Oral Tissue Regeneration & Shandong Engineering Laboratory for Dental Materials and Oral Tissue Regeneration, Shandong University, Jinan, Shandong, China
[2] School of Philosophy, Psychology and Language Sciences, College of Humanities and Social Science, The University of Edinburgh, Edinburgh, UK
[3] Department of Microbiology, Key Laboratory for Experimental Teratology of the Chinese Ministry of Education, School of Basic Medical Science, Shandong University, Jinan, Shandong, China
[4] School of Medicine, Shandong University, Jinan, Shandong, China
* These authors contributed equally to this work.

Corresponding authors
Yue Fu, fuyuesdu@163.com
Xin Xu, xinxu@sdu.edu.cn

## ABSTRACT

Oral squamous cell carcinoma (OSCC) has always been one of the most aggressive and invasive cancers among oral and maxillofacial malignancies. As the morbidity and mortality of the disease have increased year by year, the search for a promising diagnostic and prognostic biomarker for the disease is becoming increasingly urgent. Tumorous and adjacent tissues were collected from three OSCC sufferers and we obtained 229 differentially expressed genes (DEGs) between tumor and normal tissues via high-throughput RNA sequence. Function and pathway enrichment analyses for DEGs were conducted to find a correlation between tumorigenesis status and DEGs. Protein interaction network and molecular complex detection (MCODE) were constructed to detect core modules. Two modules were enriched in MCODE. The diagnostic and prognostic values of the candidate genes were analyzed, which provided evidence for the candidate genes as new tumor markers. Small Proline Rich Protein 3 (SPRR3), a potential tumor marker that may be useful for the diagnosis of OSCC, was screened out. The survival analysis showed that SPRR3 under expression predicted the poor prognosis of OSCC patients. Further experiments have also shown that the expression of SPRR3 decreased as the malignancy of OSCC increased. Therefore, we believe that SPRR3 could be used as a novel diagnostic and prognostic tumor marker.

Subjects Bioinformatics, Genomics, Dentistry, Oncology
Keywords Oral squamous cell carcinoma, Bioinformatics, SPRR3, Biomarker, High-throughput RNA sequence

## INTRODUCTION

Oral squamous cell carcinoma (OSCC) is a well-known malignancy that frequently arises from the epidermal layer in oral cavity and, in most of the cases, develops from
precancerous lesions of the oral mucosa. The majority of the OSCC patients are among those affected by head and neck squamous cell carcinoma (HNSCC), with an overall 5-year survival rate of ~64.4%. The outcome of OSCC treatment may vary depending on a series of factors, such as age, race, tumor staging, secondary complications and location of the tumor in the oral cavity (Zanoni et al., 2019). So far, current therapeutic strategies have been unable to fundamentally predict and fully cure OSCC patients. The mechanisms involved in the transformation of the normal oral epithelium into OSCC have not been fully characterized (Abdalla et al., 2017), therefore limiting any prediction on tumor progression and potential therapeutic approaches. Despite the various methods used for identifying the tumor type/stage and corresponding prognosis, most of these approaches have limited use to precisely specify the occurrence and development of the disease. Due to postoperative recurrence and lack of targeted chemical drugs, further innovations are urgently needed in regard to OSCC diagnosis and treatment. Therefore, a better prediction method for OSCC prognosis has been proposed, namely using gene expression data combined with clinical information of affected patients. Since microarray analyses and high-throughput sequencing techniques have been largely utilized in oncology, a representative number of genetic and molecular alterations have been reported (Shridhar et al., 2016). Still, the use of single markers to accurately diagnose malignant tumors is challenging. Thus, it appears critical to collectively distinguish candidate genes that are essential in cancer progression, from unrelated ones.

In general terms, Bioinformatics refers to a cutting-edge method of biological information analysis with obvious advantages when compared with traditional trials, such as the high efficiency to analyze extensive amounts of supporting data. As a result, Bioinformatic approaches have been continuously applied in cancer research to eliminate inconsistent results led by small sample size and/or by applying different technological platforms. However, the application of Bioinformatic techniques in the field of OSCC research is still restricted, despite the multitude of available information awaiting to be mined in public databases. Instead of performing isolated bioinformatics analysis, we have presently added biopsy RNA sequencing from OSCC patients during the analytical process. We further confronted this analysis with large sampling of microarray data in the current databases, accompanied with immunohistochemical (IHC) staining, therefore increasing the reliability of endpoint results. According to this Bioinformatic workflow, Small Proline Rich Protein 3 (SPRR3) was screened out as a potential prognostic and diagnostic biomarker for OSCC.

Small Proline Rich Protein 3 is a protein coding gene largely expressed in normal epithelial cells, whose gene product is known as a small proline-rich protein typically present in epidermal differentiation complexes. Importantly, SPRR3 has been recognized as a marker of terminal squamous cell differentiation (Candi, Schmidt & Melino, 2005), but its role in OSCC has been rarely studied. In our current work, we have established a bioinformatic-based workflow (Fig. 1) to screen novel diagnostic and prognostic OSCC markers as well as to verify their predictive efficiency in OSCC patients. As a result, we were able to identify SPRR3 as a putative target marker in OSCC.

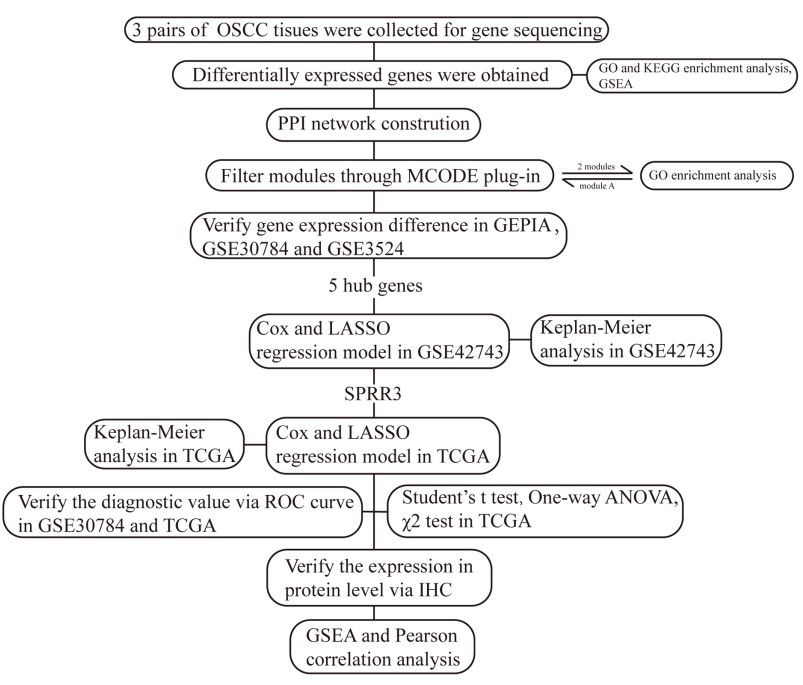

**Figure 1 The workflow of this study.**

## MATERIALS AND METHOD

### Information of specimens

We collected three pairs of tumor and adjacent oral normal specimens from three patients diagnosed with OSCC, whose diagnoses were supported by pathological examination results. All the three patients, coming from Stomatological Hospital of Shandong University, were considered as candidates for curative surgical resection (patients' details in Table S1). This study was approved by the Research Ethics Committee of Stomatological Hospital of Shandong University (No. 20190205). Tissue microarray chips consisted of 61 OSCC samples and 10 samples of normal control was constructed by Shanghai Qutdo Biotech Company (Shanghai, China. Patients' details in Table S2). The tissue microarray construction was approved by Taizhou Hospital Research Ethics Committee in Zhejiang Province (YB M-05-01). All experiments were performed complying with the relevant regulations, and written informed consent was obtained from patients.

### High-throughput RNA sequence

We performed poly-A sequencing on the three pairs of tissues. According to the manufacturer's protocol, we extracted total RNA from the three pairs of tissues we collected mentioned above by Trizol (Invitrogen, Carlsbad, CA, USA), and removed ribosomal RNA by the Ribo-Zero™ kit (Epicentre, Madison, WI, USA). The first and second strand cDNA synthesis of the fragment RNA (the average length of 200 bp) was performed, followed by adaptor ligation, and low cycle enrichment in accordance with the instructions of NEBNext® Ultra™ RNA Library Prep Kit for Illumina (New England Biolabs, Ipswich, MA, USA). In addition, the clean data quality control was filtered using
Trimmomatic (Version 3). We utilized the Agilent 2200 TapeStation (Life Technologies, Waltham, MA, USA) and Qubit®2.0 (Invitrogen, Carlsbad, CA, USA) to evaluate the purified library products. The libraries were paired-end sequenced (PE150, Sequencing reads were 150 bp) at Guangzhou RiboBio Co., Ltd. (Guangzhou, Guangdong province, China) using IlluminaHiSeq 3000 platform (NEB).

Paired-end reads alignment to the human reference genome hg19 (NCBI build 37.1, http://www.ncbi.nlm.nih.gov/projects/genome/assembly/grc/) were performed with HISAT2 (http://ccb.jhu.edu/software/hisat2/manual.shtml#getting-started-with-hisat2, *Pertea et al., 2016*). We use HTSeq v0.6.0 (*Anders, Pyl & Huber, 2015*) to count the reads numbers mapped to each gene. The expression levels were presented as RPKM (expected number of Reads PerKilobase of transcript sequence per Million base pairs sequenced), which was the recommended and most frequently-used measuring unit. And the RNA-seq data was uploaded to GEO (Gene Expression Omnibus) database (GSE140707).

## Data processing for public databsets

Gene Expression Profiling Interactive Analysis (GEPIA) (*Tang et al., 2017*) is an interactive web server for analyzing the RNA-seq expression data from the TCGA (the Cancer Genome Atlas) (https://tcga-data.nci.nih.gov/tcga/) (*Tomczak, Czerwińska & Wiznerowicz, 2015*) and Genotype-Tissue Expression (GTEx) projects. GEPIA provides customizable functions such as tumor/normal differential expression analysis, profiling according to cancer types or pathological stages, patient survival analysis, etc.

GSE3524 (*Toruner et al., 2004*), GSE42743 (*Lohavanichbutr et al., 2013*) and GSE30784 (*Chen et al., 2008*) datasets in the GEO database were selected as data source for OSCC information. Among them, GSE30784 and GSE3524 are the datasets containing gene expression information in both tumor and normal oral tissues, and GSE42743 is the one containing patients' survival and clinical data such as survival time, live status, and other related factors that may influence the development and occurrence of OSCC. All of the datasets were selected to increase the sample size of the RNA-seq microarray data, so as to better verify the differential expression and diagnose the prognosis effect of the candidate genes. The platform files and original files (.CEL files) were downloaded. With the aid of Robust Muliti-array Average (RMA) algorithm, data imputation, background correction, $log_2$ conversion, quantile normalization, pro summarization and missing value supplementation were performed on the matrix data of each GEO dataset through the "impute" package and "affy" package (*Gautier et al., 2004*) in R/Bioconductor software (*Szklarczyk et al., 2011*) (version 3.5.3). A collection of 306 specimens of OSCC was obtained from TCGA database, the level 3 RNA sequence (RNA-seq) data (.count files) was downloaded from TCGA website (https://www.cancer.gov/about-nci/organization/ccg/research/structural-genomics/tcga), and their correlated clinical information from FireBrowse database (http://firebrowse.org/) (*Chabanais et al., 2018*). The RNA-seq data in the TCGA database was normalized and processed using the "edgR" package (*Zhang et al., 2019*) prior to $log_2$ conversion (*Yang et al., 2019*).

## Differential expression analysis

All the differentially expressed genes (DEGs) were obtained through high-throughput RNA-seq. The statistically significant DEGs were acquired by utilizing the "DEseq" package (*Wang et al., 2010*), and set adjusted P-value (FDR) < 0.05 and |log2 (fold change (FC)) | > 1 as inclusion criteria. Then a hierarchical clustering analysis was performed using the "gplots" package (*Ma et al., 2017*) in R/Bioconductor software based on the expression level of DEGs in different groups, and colors represent different clustering information.

## Construction of a protein–protein interaction network and module analysis

To assess the inter-relationships among DEGs, at protein level, these were mapped using the Search Tool for the Retrieval of Interacting Genes (STRING) database (*Szklarczyk et al., 2017*). The inter-relation among DEGs (combined score ≥ 0.4) was utilized to design a protein–protein interaction (PPI) network using Cytoscape (*Shannon et al., 2003*) (v3.6.1). The molecular complex detection (MCODE) plugin was used to select meaningful modules in the PPI network. For this, MCODE $k$-core > 2 and MCODE score > 5 were utilized as cut-off values. MCODE is a widely used algorithm for predicting molecular complexes, according to the inter-connectivity and density between nodes in PPI networks. The respective score determines the number of included nodes in clusters as well as the cluster size. An appropriate score is more likely to help select the meaningful cluster, which bears potential biological functions. The $k$-core is a vertex weighting method. The weighting scheme further boosts the weight of densely connected vertices, to further optimize the module-based cluster scoring system. A higher k-core of a particular network is the central and most densely connected subnet (*Bader & Hogue, 2003*).

## Kyoto encyclopedia of genes and genomes pathway and Gene Ontology enrichment analysis

To determine the functions of respective DEGs, we initially performed a Gene Ontology (GO) enrichment analysis (*Liu, Liu & Rajapakse, 2018*) via the Database for Annotation, Visualization and Integrated Discovery (DAVID) database (Version 6.8, https://david. ncifcrf.gov/) (*Huang, Sherman & Lempicki, 2009*). The enrichment of different pathways was mapped using the Kyoto encyclopedia of genes and genomes (KEGG) pathway analysis (*Kanehisa et al., 2017*). An FDR < 0.05 was set as an inclusion criterion in both GO and KEGG enrichment analyses. In order to identify putative roles of distinct gene modules in OSCC, we performed the GO enrichment analysis for DEGs that composed each selected module cluster. The "cluster Profiler" package in R/Bioconductor software (*Yu et al., 2012*) was utilized for this enrichment analysis. The FDR < 0.05 was set again as an inclusion criterion in this GO analysis.

## IHC staining

The included patients' clinical and pathological data were shown in Table S3. The tissue microarray chips were dewaxed and dehydrated, and then incubated overnight at 4 °C with

monoclonal rabbit anti-human SPRR3 (Abcam, ab218131) after epitope retrieval, $H_2O_2$ treatment and non-specific antigens blocking, after signals detection by using DAB staining kit (Vector Laboratories, Burlingame, CA, USA), we incubated chips with the secondary antibody. IHC analysis of the chips were performed under optical microscopes of 100× and 400×.

## Gene-set enrichment analysis

Gene-set enrichment analysis (GSEA) is a computer-based tool used for microarray data analysis and annotation, which is based on biological knowledge (*Subramanian et al., 2005*). The GSEA enrichment score (ES) reflects the degree to which a particular set is over-represented at the extremes (top or bottom) of the entire ranking list. ES is normalized for each geneset to account for the overall size of the set, yielding a normalized enrichment score (NES). In this study, the data of OSCC was respectively obtained from our own RNA-seq microarray data and TCGA database. RNA-seq samples were grouped as tumoral and normal tissues. Samples from TCGA database were grouped according to the quantile of SPRR3 expression levels. Thereafter, samples were sorted from low to high in terms of expression level, where the first quantile was defined as the "low expression group" while the last quantile was defined as the "high expression group". GSEA annotated the GO, KEGG and Hallmark enrichment results of these genes with an enrichment score. The inclusion criteria of the enriched results were established as |NES| > 1, *P*-value < 0.05 and FDR < 0.25 (*Subramanian et al., 2005*).

## Statistical analysis

To explore the connection between factors and overall survival (OS) of the OSCC patients, univariate Cox regression model was utilized in specimens of GSE42743. Subsequently, we used significant factors to conduct the Least Absolute Shrinkage and Selection Operator (LASSO) method. LASSO regression contributes to variable selection and regularization, at the same time that it fits the generalized linear model. Using this statistical approach, we could also avoid over-fitting at some extent. Because the complexity of LASSO regression was controlled by the coefficient (λ), a less variable model was obtained by performing a penalty ratio, according to their size. A relatively small number of highly correlated indicators was then obtained and, subsequently, a plot was designed according to the partial possibility deviance vs log(λ). The positive factors of the LASSO regression analysis were incorporated into the multivariant Cox regression model, which confirmed the feasibility of these factors as independent predictors of OSCC. We later verified the effect(s) in the TCGA database using this same methodology.

The K–M survival analysis was performed to examine the relationship between the survival time of OSCC patients and the expression level of respective DEGs. OSCC samples from GSE42743 and TCGA were divided into two groups, according to the quartile of expression for each selected gene. As previously, samples were sorted from low to high in terms of expression level for each target gene. First and last quantiles were related to the low and high expression groups, respectively. We calculated and then indicated the hazard ratio (HR) with 95% confidence intervals and log-rank *P*-value along the plot.

To evaluate the discriminatory accuracy of a selected gene between two groups, its expression level was included in the Receiver Operating Characteristic (ROC) curve analysis. The ROC curve is an essential tool for diagnostic detection or biomarker prediction in a certain disease (*Matthews, Ranson & Kelso, 2011*). Plots of sensitivity (true-positive fraction) versus 1-specificity (false-positive fraction) were constructed. The value for the area under ROC curve (AUC) corresponds to the ability of one gene to distinguish tumor tissues from adjacent ones. $\chi^2$ test was utilized to verify the correlation between gene expression levels and clinicopathologic factors. In this context, samples were also grouped according to the quantile of expression level of a respective target gene. All statistical analyses were conducted by SPSS 25.0 software (SPSS Inc., Chicago, IL, USA). The correlation coefficient between two genes was calculated by Pearson correlation analysis. For expression data comparison, student's *t* test (two-tailed) and ANOVA were performed using GraphPad Prism 8 software.

# RESULTS

## DEGs search and analysis

According to the sequencing data here available, a total of 229 DEGs were identified in OSCC samples, from which 85 genes were up-regulated while 144 were down-regulated. The most common up- ($\log_2FC > 1$, FDR < 0.05) and down-regulated ($\log_2FC < -1$, FDR < 0.05) genes are listed in Table S4. Both volcano plot and heatmap were generated to show the discrepancy among the expressed genes when comparing OSCC and adjacent normal tissues (Figs. 2A and 2B).

## The enrichment results of the DEGs

Based on the GO enrichment analysis, the commonly enriched categories were (i) keratinization, (ii) keratinocyte differentiation, (iii) cell adhesion, (iv) migration and (v) proliferation. In addition, DEGs were enriched in alcohol and drug metabolic processes (Fig. 2C), which have been closely related to both tumorigenesis and development. The most stringent results of GO enrichment analysis, including BP (biological process), MF (molecular function) and CC (cellular component) terms, (the top 100 items of each terms with the lowest FDR) are presented in Table S5.

Kyoto encyclopedia of genes and genomes pathway analysis showed that the majority of DEGs were enriched in (i) drug metabolism, (ii) retinol metabolism, (iii) chemical carcinogenesis and (iv) leukocyte transendothelial migration pathways (Fig. 2D). Detailed results of KEGG enrichment analysis are presented in Table S6.

In order to validate the DEG-related functions and pathways on an unbiased basis, GSEA analysis was performed using previously generated RNA-seq data. DEGs were mostly enriched in (i) epithelial mesenchymal transition (EMT), (ii) fatty acid metabolism, (iii) K-Ras signaling down and (iv) TGF-β signaling. In regard to the GO-BP terms, GSEA showed enrichment of DEGs in (i) DNA replication, (ii) epidermal cell differentiation and (iii) positive regulation of cell activation. In the context of KEGG enrichment, DEGs were largely present in (i) drug metabolic cytochrome P450, (ii) extracellular matrix receptor interaction, (iii) focal adhesion and (iv) tight junction

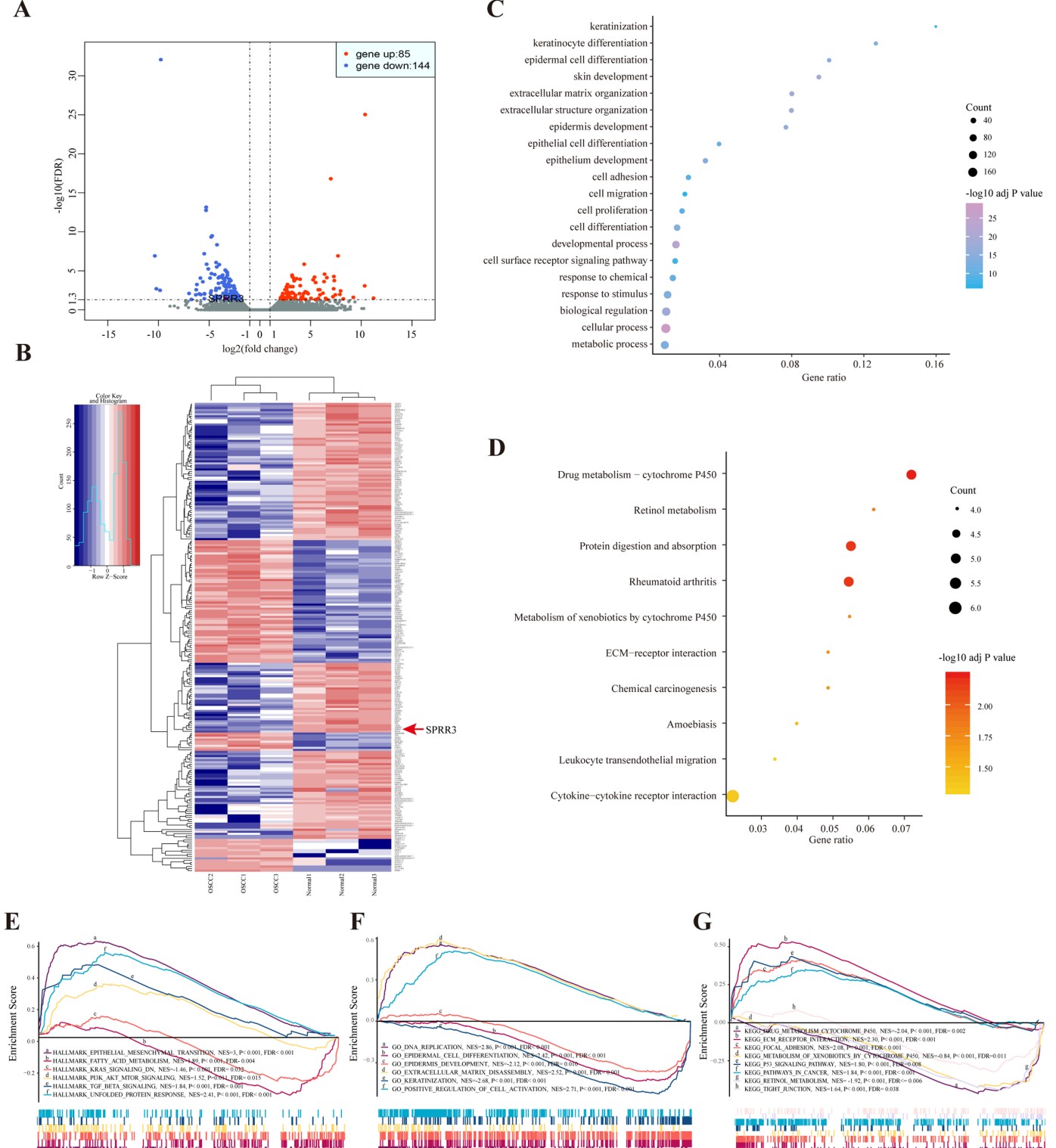

**Figure 2 DEGs between OSCC and adjacent tissues.** (A) Volcano plot exhibits the high-throughput RNA-seq result. Among them, 85 genes were up-regulated (red plots) and 144 were down-regulated (green plots). SPRR3 was marked in the figure. (B) Heatmap of RNA sequencing result, red for up-regulated genes and blue for down-regulated genes. SPRR3 was marked in the figure. (C) GO enrichment of the DEGs. The DEGs are mostly correlated with keratinization, keratinocyte differentiation, ECM organization, cell adhesion and migration. (D) KEGG enrichment of the DEGs. The DEGs were enriched in xenobiotic metabolism, drug metabolism and chemical carcinogenesis, etc. (E) GSEA in hallmark enrichment, NES,
**Figure 2** (continued)
*P*-value and FDR were listed respectively. The DEGs were enriched in epithelial mesenchymal transition, K-Ras signaling down and TGF-β signaling, etc. (F) GSEA in GO enrichment. The DEGs were enriched in DNA repication, epidermal cell differentiation, ECM disassembly and positve regulation of cell activation, etc. (G) GSEA in KEGG enrichment. The DEGs were enriched in drug metabolism, xenobiotic metabolism, focal adhesion, P53 signaling and tight junction, etc.                     

(Figs. 2E–2G). Above all, DEGs were more concentrated in biological processes including (i) epidermal cell differentiation, (ii) metabolic process and (iii) cell adhesion, reiterating a potential correlation with tumorigenesis and cancer development.

## Identification of module clusters via PPI network

A global PPI network was designed after establishing the inter-relationships among the annotated DEGs (STRING database search). As a result, 166 nodes with a combined score > 0.4 were obtained (Fig. S1). MCODE plugin was applied for module analysis, where two modules were chosen for further analysis according to the inter-connectivity and density between nodes in PPI networks. Module clusters 1 and 2 consisted, respectively, of eight and ten nodes (Figs. 3A and 3B). All DEGs present in these two particular modules are shown in Table 1.

Genes from selected module clusters were included in the GO enrichment analysis to properly understand their functions and gene product attributes. In regard to BP, the DEGs from module cluster 1 were correlated with (i) keratinocyte differentiation, (ii) epidermal cell differentiation and (iii) epidermis development. In terms of CC, the DEGs were enriched in cornified envelope and desmosomes, and for MF, the DEGs were enriched in protein binding, bridging and molecular adaptor activity (Figs. 3C–3E). The DEGs in module cluster 2 were mainly correlated with collagen metabolic process and extracellular matrix organization (Figs. 3F–3E). Interestingly, most of the genes here involved (for instance, Matrix Metallopeptidase (MMP) family members) have been associated with tumor invasion and metastasis (*Lu et al., 2017*; *Zou et al., 2019*). The MCODE score of module cluster 1 was higher, and the hub genes in this cluster were barely reported in OSCC studies. Hence, the module cluster 1 was preliminarily chosen for subsequent evaluation.

## Validation of DEGs in independent OSCC datasets

According to the TCGA database, OSCC accounts for the majority of HNSCC cases. The RNA expression level of DEGs in module cluster 1 was preliminarily analyzed by GEPIA in the HNSCC dataset. Among these DEGs, CSTA (Cystatin A), EVPL (Envoplakin), PPL (Periplakin), SPRR3, TGM1 (Transglutaminase 1) were significantly down-regulated in tumor tissues when compared with their normal counterparts (*P* < 0.05) (Figs. 4A–4H). To further demonstrate that these five genes are differentially expressed in OSCC, we utilized the GSE30784 and GSE3524 datasets to confirm potential disease-related DEGs. Consistently, these five genes were significantly down regulated in tumor tissues (Figs. 4I–4M for GSE30784; Figs. S2A–S2F for GSE3524).

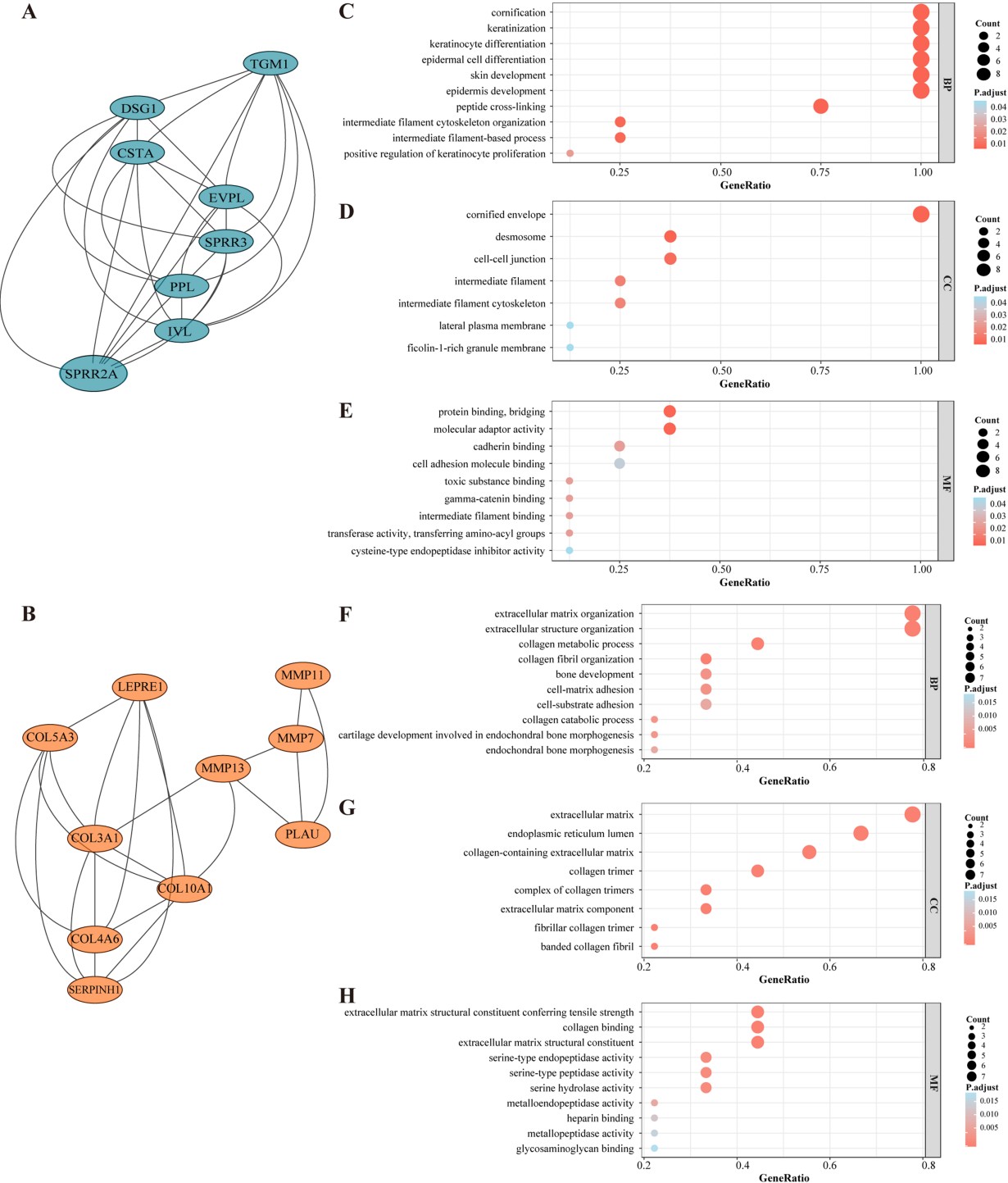

**Figure 3 Selected core modules of PPI network and functional enrichment analysis.** (A) Genes in module cluster 1 (blue nodes) were down-regulated in OSCC tissues. (B) Genes in module cluster 2 (red nodes) were up-regulated in OSCC. (C) GO enrichment shows that module cluster 1 was correlated with keratinization, cornification and epidermal cell differentiation in BP. (D)Genes in molecular cluster 1 were enriched in cornified envelop, desmosome and cell junction etc. in CC. (E) Genes in molecular cluster 1 were enriched in protein binding and bridging in MF. The size of the dot refers to the gene counts enriched in that term, and the color refers to the FDR according to the FDR gauge. (F) Module cluster 2 was mostly enriched in collagen metabolic process and extracellular structure organization in BP. (G) Genes in molecular cluster 1 were enriched in extracellular matrix in CC. (H) Genes in molecular cluster 1 were enriched in collagen binding in MF.

**Table 1 The DEGs in the two module clusters screened out by MCODE score.**

| MCODE cluster | MCODE score | Gene ID |
|---|---|---|
| Cluster 1 | 7 | SPRR2A |
| Cluster 1 | 7 | EVPL |
| Cluster 1 | 7 | PPL |
| Cluster 1 | 7 | DSG1 |
| Cluster 1 | 7 | CSTA |
| Cluster 1 | 7 | TGM1 |
| Cluster 1 | 7 | SPRR3 |
| Cluster 1 | 7 | IVL |
| Cluster 2 | 5 | COL4A6 |
| Cluster 2 | 5 | MMP13 |
| Cluster 2 | 5 | LEPRE1 |
| Cluster 2 | 5 | SERPINH1 |
| Cluster 2 | 5 | COL3A1 |
| Cluster 2 | 5 | COL10A1 |
| Cluster 2 | 5 | COL5A3 |
| Cluster 2 | 5 | MMP7 |
| Cluster 2 | 5 | PLAU |

## Identification of DEG-related prognostic value

To properly select the predictive factors associated with the overall survival of OSCC patients, the five above-mentioned genes and clinical disease characteristics were introduced into univariate Cox regression model using the GSE42743 dataset. The most statistically significant DEGs (SPRR3, PPL, TGM1) and N staging were concomitantly examined by LASSO method. As a result, only SPRR3 and N staging were confidently sorted out (Figs. 5A and 5B). To further determine whether low SPRR3 expression could be an independent predictor of OSCC prognosis, a multivariate Cox regression model was executed. As predicted, multivariate Cox regression analysis showed that SPRR3 expression levels (HR = 0.865, 95% CI [0.754–0.992], $P$-value = 0.037) act as an independent prognostic factor for the OS of OSCC patients (Table 2).

The SPRR protein family has been reported to be functional in a variety of tumors (*Specht et al., 2013*). Still, the expression and function of SPRR3 in OSCC remain largely unclear. Thus, we have presently proposed SPRR3 as an OSCC-related gene and attempted to explore its role in tumorigenesis and development of this condition. Therefore, Kaplan–Meier (K–M) survival analysis was carried out using the GSE42743 dataset to verify whether SPRR3 could act as a prognostic marker (log rank $P$-value < 0.05; Fig. 5C).

Statistical platforms including univariate Cox regression model, LASSO method and Cox multivariate regression model were also approached to validate the SPRR3's prognostic value, based on the TCGA database. According to the univariate Cox analysis, factors including low SPRR3 expression, complete tumor remission, lymphovascular and perineural invasion, as well as N staging were significantly associated with a poor

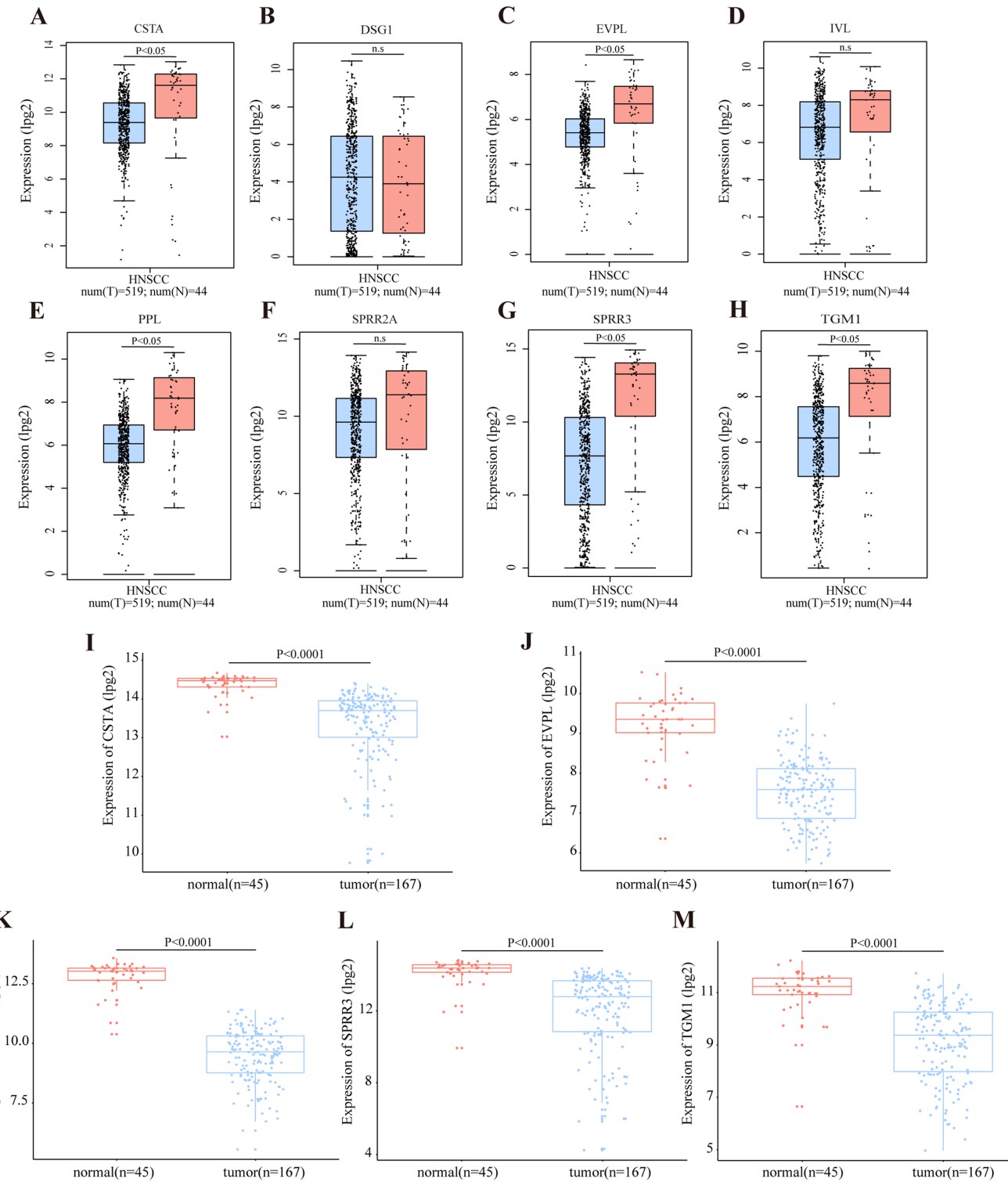

**Figure 4 Transcriptional level of DEGs in OSCC.** (A–H) The transcription level of DEGs in module cluster 1 according to GEPIA. The expression levels of CSTA, PPL, EVPL, SPRR3 and TGM1 with *P*-value < 0.05 in student's *t* test (two tailed) were significantly differentially expressed between tumor and normal tissues in HNSCC (n.s represents not significant). (I–M) The differentially transcript level of candidate genes in module cluster 1 in GSE30784 dataset. The expression levels of CSTA, PPL, EVPL, SPRR3 and TGM1 with *P*-value < 0.001 were significantly differentially expressed between tumor and normal tissues in OSCC by student's *t* test (two tailed).

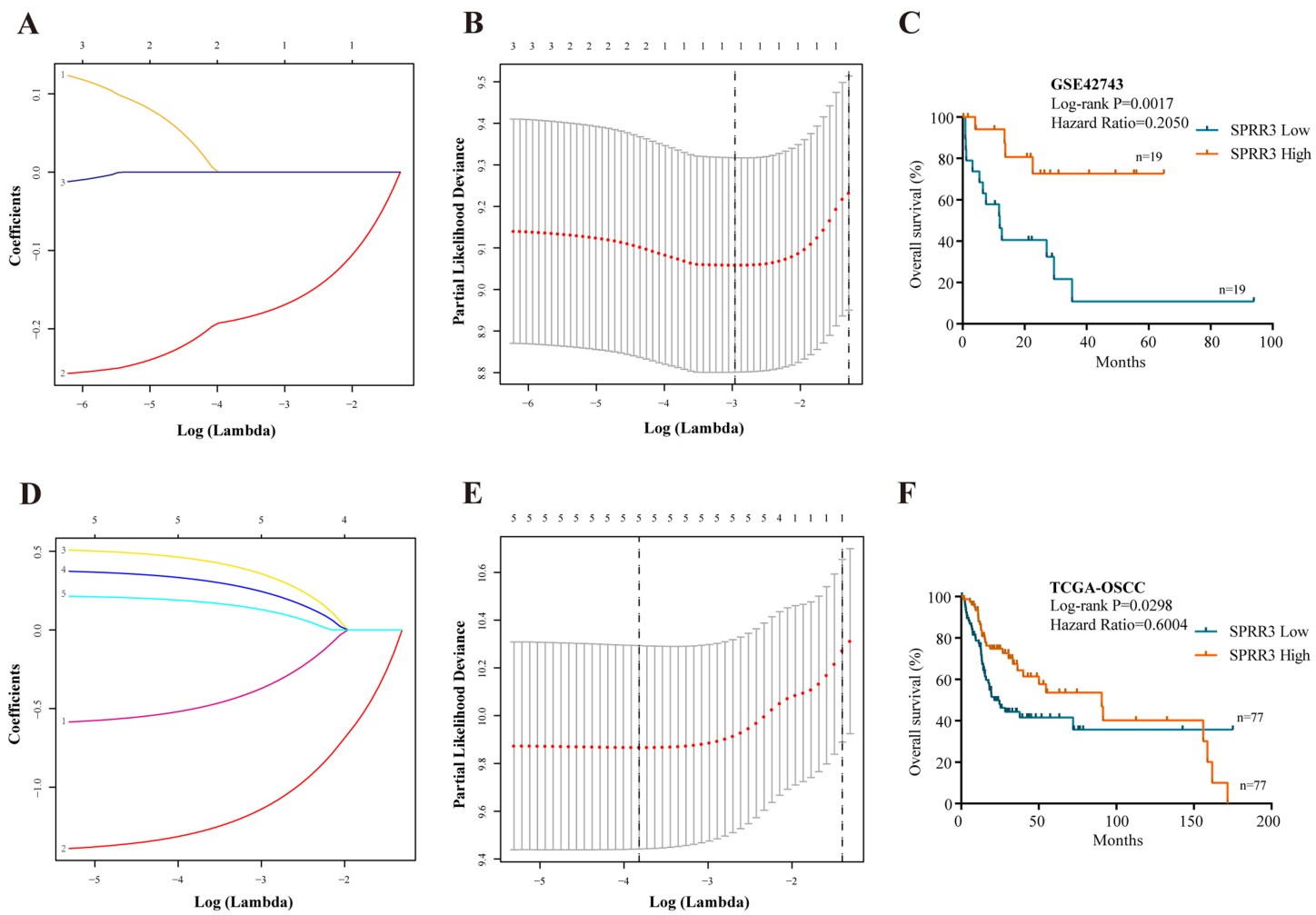

**Figure 5 Prognostic significance.** (A) Validation was performed for tuning parameter selection through the LASSO regression analysis in GSE42743. LASSO, the least absolute shrinkage and selection operator. (B) Elaboration for LASSO coefficient profiles of prognostic RNAs. (C) K–M survival analysis based on GSE42743 dataset according to SPRR3 expression level. (D) Validation was performed for tuning parameter selection through the LASSO regression model in TCGA. (E) Elaboration for LASSO coefficient profiles of prognostic RNAs. (F) K–M survival analysis based on TCGA-OSCC dataset according to SPRR3 expression level.

prognosis. SPRR3 was also screened out by the LASSO method (Figs. 5D and 5E). Multivariate Cox proportional regression analysis revealed that SPRR3 expression levels (HR = 0.544, 95% CI [0.329–0.899], $P$-value = 0.017) was an independent prognostic factor for the OS of OSCC patients (Table 3). The predictive value of SPRR3 towards OSCC prognosis was confirmed by K–M survival analysis based on TCGA database (log rank $P$-value < 0.05; Fig. 5F).

## Low SPRR3 expression is correlated with carcinogenesis and progression of OSCC

According to the OSCC patient data in the TCGA database, SPRR3 expression levels were significantly lower than those in oral normal tissues ($P$-value < 0.0001, Fig. 6A). ROC analysis was then performed to confirm the statistical relevance of these results. For this,

**Table 2 Cox and LASSO regression analysis for OS in patients with OSCC from GSE42743.**

| Characteristics | Univariate analysis | | LASSO | Multivariate analysis | |
|---|---|---|---|---|---|
| | HR (95% CI) | *P*-value | | HR (95% CI) | *P*-value |
| SPRR3 expression High expression | 0.801 [0.662–0.969] | **0.032** | (+) | 0.865 [0.754–0.992] | **0.037** |
| CSTA expression High expression | 0.924 [0.759–1.126] | 0.435 | | | |
| EVPL expression High expression | 0.741 [0.480–1.146] | 0.178 | | | |
| PPL expression High expression | 0.771 [0.616–0.965] | **0.023** | (−) | | |
| TGM1 expression High expression | 0.801 [0.662–0.969] | **0.022** | (−) | | |
| Age (years) $\geq 60$ | 1.001 [0.978–1.025] | 0.920 | | | |
| Gender Male | 0.694 [0.347–1.386] | 0.300 | | | |
| Smoking Current | 1.032 [0.686–1.552] | 0.881 | | | |
| T stage $T_{3-4}$ | 1.39 [0.978–1.965] | 0.067 | | | |
| N stage Non $N_0$ | 1.970 [1.370–2.832] | **<0.001** | (+) | 1.750 [1.202–2.548] | **0.003** |

Note:
T, primary tumor; N, regional lymph nodes; (+), selected by LASSO regression; (−), abandoned by LASSO regression. The bold characters refers to the *P*-values of indicators that were statistically significant in the regression models.

AUC was calculated to identify the diagnostic specificity and sensitivity of SPRR3 expression. Based on the GSE30784 dataset and the TCGA database, down regulated SPRR3 levels yielded an AUC of 0.920 and 0.731, respectively (Figs. 6B and 6C). In this regard, SPRR3 had positive diagnostic performance for OSCC patients' discrimination. Thus, SPRR3 could act as a potential diagnostic indicator for OSCC.

Student's *t*-test (two-tailed) and one-way ANOVA were thus performed and showed that low SPRR3 expression was in the subgroups of patients with high alcohol consumption, poor differentiation, non $N_0$ staging, positive lymphovascular invasion, and positive perineural invasion (Figs. 6D–6H). The $\chi^2$ test was also utilized to figure out the correlation between SPRR3 expression of and the clinicopathological characteristics of OSCC. Consistent with previous tests, SPRR3 expression was significantly correlated with alcohol consumption (*P*-value = 0.011), histologic grade (*P*-value < 0.001), N staging (*P*-value = 0.005), lymphovascular invasion (*P*-value = 0.026) and perineural invasion (*P*-value = 0.016) (Table 4). These results reiterated that SPRR3 may act as a factor related to tumor progression and/or metastasis.

## Differential detection of SPRR3 protein in OSCC

Immunohistochemical staining suggests that the SPRR3 protein levels in OSCC tissues vary according to the histologic grade of the cancer (Figs. 6I–6L). Specifically, SPRR3 levels

**Table 3 Cox and LASSO regression analysis for OS in patients with OSCC from TCGA.**

| Characteristics | Univariate analysis | | LASSO | Multivariate analysis | |
|---|---|---|---|---|---|
| | HR (95% CI) | P-value | | HR (95% CI) | P-value |
| SPRR3 expression High expression | 0.595 [0.371–0.955] | **0.032** | (+) | 0.544 [0.329–0.899] | **0.017** |
| Age (years) ≥60 yr | 1.086 [0.676–1.745] | 0.734 | | | |
| Gender Male | 0.788 [0.448–1.272] | 0.329 | | | |
| Alcohol history Yes | 1.363 [0.806–2.305] | 0.247 | | | |
| HPV infection Positive | 1.138 [0.518–2.497] | 0.748 | | | |
| Radiation therapy Yes | 0.639 [0.398–1.025] | 0.063 | | | |
| Histologic grade Moderate and Poor | 1.549 [0.760–3.156] | 0.228 | | | |
| Tumor stage III–IV | 1.558 [0.898–2.704] | 0.115 | | | |
| T stage $T_{3-4}$ | 1.607 [0.977–2.643] | 0.062 | | | |
| Primary therapy outcome Complete remission | 0.282 [0.176–0.453] | **<0.001** | (+) | 0.242 [0.147–0.397] | **<0.001** |
| Lymphovascular invasion Yes | 2.109 [1.266–3.512] | **0.004** | (+) | 1.690 [0.991–2.881] | 0.054 |
| Perineural invasion Yes | 1.727 [1.081–2.759] | **0.022** | (+) | 1.475 [0.857–2.538] | 0.161 |
| N stage Non $N_0$ | 1.833 [1.131–2.971] | **0.014** | (+) | 1.250 [0.711–2.194] | 0.438 |

Note:
T, primary tumor; N, regional lymph nodes; (+), selected by LASSO regression. The bold characters refers to the *P*-values of indicators that were statistically significant in the regression models.

were lower in tumor tissues from patients at higher histologic grade when compared to those at lower grade.

## SPRR3 is enriched in various metabolic pathways and negatively associated with OSCC malignant progression

To probe SPRR3-related pathways in an unbiased manner, GSEA analysis was conducted using OSCC data from TCGA. Based on the hallmark signature, SPRR3 was enriched in (i) apical surface, (ii) fatty acid metabolism, (iii) K-Ras signaling down and (iv) xenobiotic metabolism. GO-BP enrichment showed that SPRR3 was also correlated with (i) alcohol metabolic process, (ii) drug metabolic process, (iii) epidermis development and (iv) fatty acid derivative metabolic process. Lastly, KEGG enrichment assigned SPRR3 into (i) drug metabolism cytochrome P450, (ii) glycolysis/gluconeogenesis, (iii) metabolism of xenobiotic by cytochrome P450, (iv) retinol metabolism and (v) VEGF signaling pathway (Figs. 7A–7C).

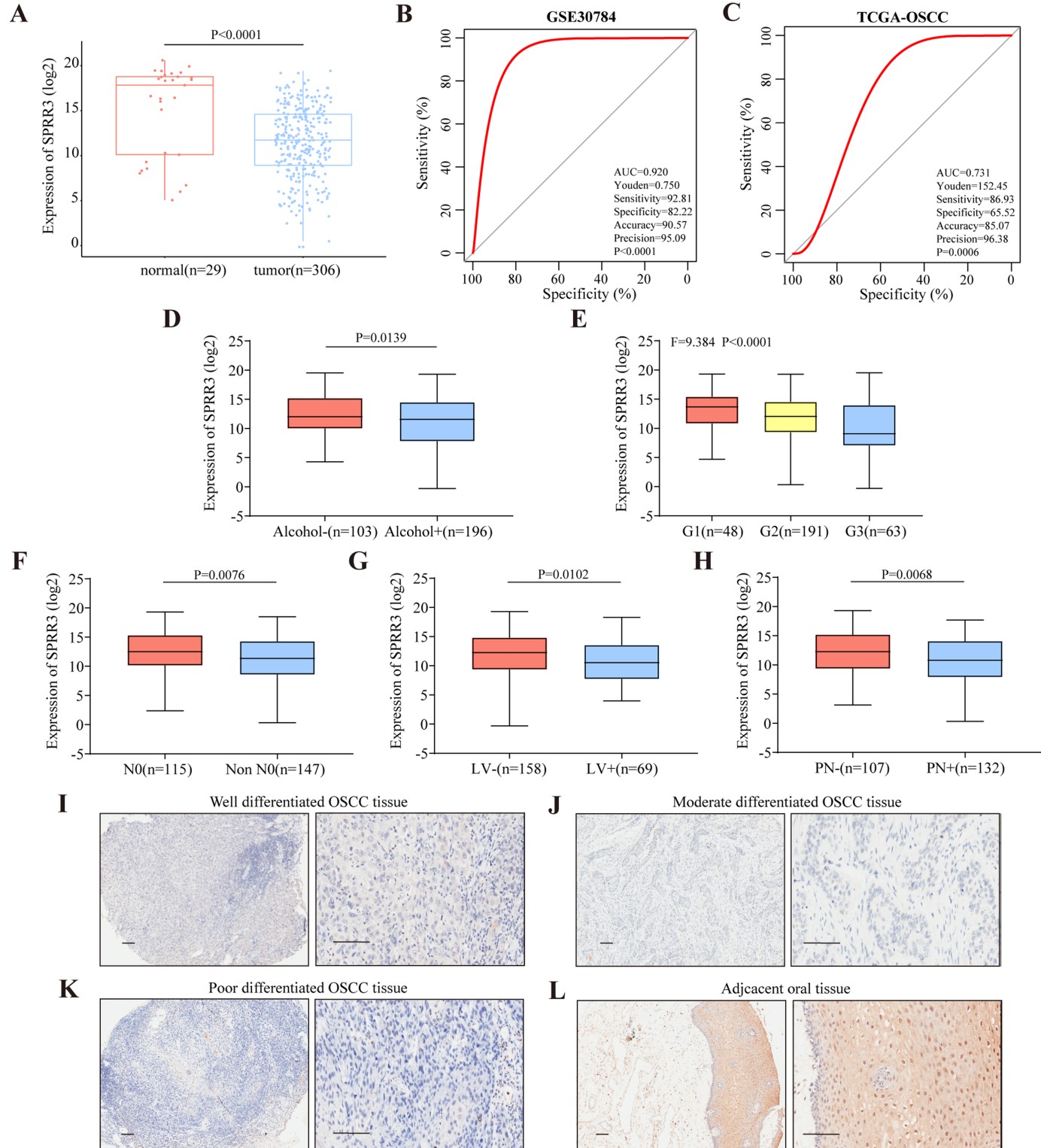

**Figure 6 Clinical correlation of SPRR3 expression level in OSCC.** (A) The expression of SPRR3 in TCGA-OSCC database, the expression level of SPRR3 was significantly lower in tumor group than that in normal group by student's *t* test (two tailed). (B) In GSE30784, ROC curve was created to verify the diagnostic value of SPRR3. AUC = 0.920, *P*-value < 0.001. (C) In ROC curve analysis from TCGA database, AUC was 0.731 and *P*-value = 0.001. (D–H) Two-tailed student's *t* test and one-way ANOVA test suggested that SPRR3 was differentially expressed in groups in accordance of whether the patients had alcohol consumption (*P* = 0.0139), the histological grade (*P*-value < 0.0001), N stage (*P*-value = 0.0076) of the

**Figure 6 (continued)**
patients, and whether the patients had lymphovascular invasion (*P*-value = 0.0102), perineural invasion (*P*-value = 0.0068) respectively. (I) Protein expression level of SPRR3 in well differentiated OSCC tissue. (J) Protein expression level of SPRR3 in moderate differentiated OSCC tissue. (K) Protein expression level of SPRR3 in poor differentiated OSCC tissue. (L) Protein expression level of SPRR3 in adjacent tissue. The image on the left in the group was taken at the low power field (100*), the right was taken at the high-power field (400*). The black line represents the length of 100 nm.

**Table 4 The relationship between the expression of SPRR3 and multiple clinicopathological factors.**

| Characteristics | | SPRR3low | SPRR3high | *P*-value |
|---|---|---|---|---|
| Gender | Female | 23 | 28 | 0.390 |
| | Male | 53 | 50 | |
| Age (years) | <60 yr | 31 | 35 | 0.515 |
| | ≥60 yr | 46 | 42 | |
| Alcohol history | No | 14 | 59 | **0.011** |
| | Yes | 59 | 47 | |
| HPV infected | Not infected | 70 | 68 | 0.575 |
| | infected | 6 | 8 | |
| Radiation | No | 30 | 30 | 1.000 |
| | Yes | 40 | 40 | |
| T stage | $T_{1-2}$ | 27 | 35 | 0.403 |
| | $T_{3-4}$ | 40 | 39 | |
| N stage | $N_0$ | 18 | 39 | **0.005** |
| | Non $N_0$ | 40 | 31 | |
| Histologic grade | G1 | 3 | 16 | **<0.001** |
| | G2 | 41 | 47 | |
| | G3 | 31 | 14 | |
| Tumor stage | I–II | 14 | 52 | 0.270 |
| | III–IV | 22 | 53 | |
| Primary therapy outcome | Progressive disease | 16 | 51 | 0.153 |
| | Complete remission | 9 | 55 | |
| Perineural invasion | No | 22 | 42 | **0.016** |
| | yes | 32 | 25 | |
| Lymphovascular invasion | No | 34 | 24 | **0.026** |
| | yes | 43 | 12 | |

**Note:**
T, primary tumor; N, regional lymph nodes. The bold characters refers to the *P*-values of indicators that were statistically significant in $\chi^2$ test.

To further verify the role of SPRR3 in OSCC carcinogenesis and development, we analyzed the relationship between SPRR3 and other genes from the two module clusters previously screened, using the GSE30784 and TCGA-OSCC datasets. Interestingly, SPRR3 was positively correlated with genes from the module cluster 1 (CSTA, DSG1, EVPL, IVL, PPL, TGM1). CSTA has been reported as a potential OSCC biomarker (*Hsiao et al., 2017*). In contrast, SPRR3 was negatively correlated with a number of DEGs from the module cluster 2, including PLAU (Plasminogen Activator, Urokinase), COL5A3

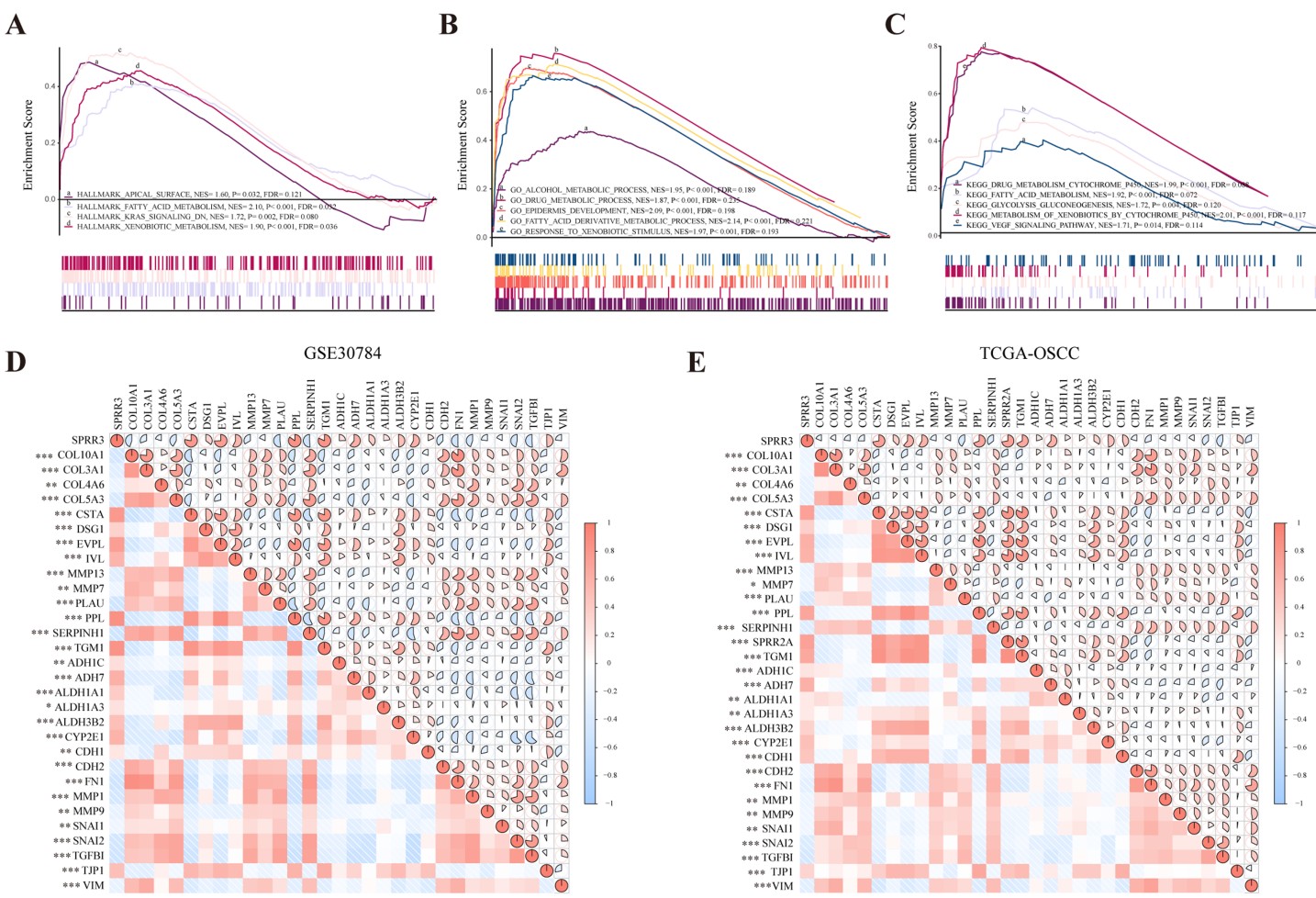

**Figure 7 GSEA analysis of SPRR3 in TCGA database.** (A) SPRR3 was enriched in apical surface, fatty acid metabolism, K-Ras signaling down and xenobiotic metabolism in hallmark terms.(B) SPRR3 was enriched in alcohol metabolic, drug metabolic, epidermis development, fatty acid derivative process and response to xenobiotic stimulus in GO terms. (C) SPRR3 was enriched in drug metabolism cytochrome P450, fatty acid metabolism, glycolysis gluconeogenesis, metabolism of xenobiotics by cytochrome and VEGF signaling pathway in KEGG terms. (D) Pearson correlation analysis between DEGs in GSE30784 dataset. SPRR3 was positively correlated with other DEGs in module cluster 1 (CSTA, DSG1, EVPL, IVL, PPL, TGM1) and negatively correlated with DEGs in Module cluster 2 (PLAU, COL10A1, COL3A1, COL4A6, COL5A3), and it was positively correlated with ADH1C, ADH7, ALDH1A1, ALDH1A3, ALDH3B2, CYP2E1, CDH1, TJP1, it was negatively correlated with CDH2, FN1, SNAI1, SNAI2 and TGFBI. (E) Pearson correlation analysis between DEGs in TCGA-OSCC dataset. SPRR3 was positively correlated with other DEGs in module cluster 1 and negatively correlated with DEGs in Module cluster 2, and it was positively correlated with ADH1C, ADH7, ALDH1A1, ALDH1A3, ALDH3B2, CYP2E1, CDH1, TJP1, it was negatively correlated with CDH2, FN1, SNAI1, SNAI2 and TGFBI. The gauge on the right refers to the "r" of Pearson correlation test. Asterisk (*), double asterisk (**) and triple asterisk (***) stand for *P*-value < 0.05, *P*-value < 0.01 and *P*-value < 0.001, respectively.

(Collagen Type V Alpha 3 Chain), SERPINH1 (Serpin Family H Member 1), MMP 13, MMP7, COL4A6 (Collagen Type IV Alpha 6 Chain), and COL3A1 (Collagen Type III Alpha 1 Chain). Coincidentally, PLAU has been reported as an important factor involved in the regulation of cell migration, proliferation and invasion in OSCC (*Zou et al., 2019*), while MMP7 appears to have an oncogenic role in OSCC invasion and metastasis (*Dasgupta et al., 2012*).

By accessing the GSE30784 and TCGA-OSCC datasets, we also examined major SPRR3 co-expressing genes, including potential markers, which were closely associated with

metabolic processes and cancer progression. As a result, we observed that SPRR3 was positively correlated with some metabolic process markers including ADH1C (Alcohol Dehydrogenase 1C), ADH7 (Alcohol Dehydrogenase 7), ALDH1A1 (Aldehyde Dehydrogenase 1 Family Member A1), ALDH1A3 (Aldehyde Dehydrogenase 1 Family Member A3), CYP2E1 (Cytochrome P450 Family 2 Subfamily E Member 1). Among the EMT- and metastasis-related markers, we have also verified that SPRR3 was positively correlated with CDH1 (Cadherin 1) and TJP1 (Tight Junction Protein 1). Moreover, SPRR3 was negatively associated with CDH2 (Cadherin 2), FN1 (Fibronectin 1), MMP1, MMP9, SNAI1 (Snail Family Transcriptional Repressor 1), SNAI2 (Snail Family Transcriptional Repressor 2), TGFBI (Transforming Growth Factor Beta Induced) and VIM (Vimentin) (Fig. 7D for GSE30784, Fig. 7E for TCGA). These results indicated that SPRR3 could be involved in a series of metabolic processes, and the downregulation of SPRR3 may be potentially linked to the malignant progression of OSCC.

## DISCUSSION

Oral squamous cell carcinoma, a subset of HNSCC, is one of the most common malignancies from the oral cavity worldwide. It is an aggressive tumor that bears a poor prognosis. In fact, OSCC patients at advanced stages, even after intensive treatment, may develop relapses and distant metastases. Therefore, early diagnosis and intervention remain critical for the management of OSCC. In this regard, measuring tumor markers has gradually become an important method for detecting the tumor status and monitoring therapeutic effects in the clinic.

The development and increasing use of tumor biomarkers has become a pivotal part of cancer diagnosis. Likewise, seeking novel tumor-related markers has provided new trains of thought in OSCC treatment (Csosz et al., 2017). Potential biomarkers with high specificity and sensitivity, identified in datasets by distinct Bioinformatic methods, have been continuously explored (Sun et al., 2015). Previous studies have determined a number of target genes by filtering DEGs in datasets (Yang et al., 2017; Zhao et al., 2018). Still, given the differences in specimens and database platforms, potential markers have failed to bear a significant clinical application value, so their verification process have been far from clinical practice. Hence, we expect that DEGs obtained from biopsy-based sequencing, combined with the validation by independent public datasets and tissues analyses, will be more beneficial for the clinical diagnosis and prognosis of the disease. In this study, we have focused on the identification of potential biomarkers which may possess diagnostic and prognostic significance.

Current RNA-seq data has revealed 229 OSCC-related DEGs that were particularly enriched in processes associated with tumorigenesis and malignant progression. By further screening gene modules in a PPI network, a module cluster containing eight down regulated hub genes in OSCC (cluster 1) was selected as a target. The core gene in this module was retrieved by acquiring OS information from different datasets. Univariate Cox regression, LASSO regression and multivariant Cox regression analyses were successively conducted in order to screen the key genes that could independently predict OSCC prognosis, resulting in the identification of SPRR3 as a central hub gene.

Thereafter, we conducted K–M survival analysis and further established ROC curves. As a predictive marker, SPRR3 has considerable clinical significance in OSCC diagnosis and prognosis. Based on clinical data, a low expression of SPRR3 was correlated with patients' alcohol consumption, histologic grade, lymph node metastasis, lymphatic vascular invasion, and perineural metastasis. GSEA and Pearson correlation analyses have been also conducted for gene annotation. Putting together, we concluded that SPRR3 is potentially associated with epidermis development, metabolic processes and tumor progression.

SPRRs are located in an evolutionarily conserved genetic cluster, designated as the epidermal differentiation complex (EDC) in the chromosome 1q21. Epidermis development and epithelial cell differentiation are some of the basic functions of SPRR proteins (*Cabral et al., 2001*; *Candi et al., 2000*). In addition, SPRRs family proteins have been reported to be active during the progression of many cancers. In OSCC, over-expressed SPRR2A (Small Proline Rich Protein 2A) can impair distant metastasis of tongue squamous cell carcinoma cells in vivo (*Fang et al., 2016*). Down-regulation of SPRR3 expression has been reported in esophageal squamous cell carcinoma, esophageal adenocarcinoma, anal and gastric cancers, while it has been strongly upregulated in colorectal and breast cancers (*Cho et al., 2010*; *Kim et al., 2012*). Nevertheless, the role of SPRR3 in oral tissues has never been explored. According to $\chi^2$ tests, here we observed that low SPRR3 expression is associated with OSCC progression, lymphovascular invasion and perineural invasion. GSEA enrichment analysis showed that SPRR3 is also correlated with K-Ras and VEGF signaling, which are classic pathways in OSCC carcinogenesis and progression (*Caulin et al., 2004*; *Siriwardena et al., 2018*).

TJP1 and CDH1 are classic inhibitors of tumor invasion and metastasis (*Li et al., 2016*). The positive correlation between SPRR3 and these two target genes indicate that SPRR3 might be essential for cell adhesion. Furthermore, SPRR3 was negatively correlated with some mesenchymal markers, such as CDH2, TGFBI, FN1, VIM, MMPs, and transcriptional factors such as SNAI1, SNAI2 (*Li et al., 2016*). Therefore, SPRR3 appears to work as a modulating factor involved in the maintenance of oral epithelial tissues and also in EMT inhibition. These particular roles may explain why patients under-expressing SPRR3 are expected to have a relatively poor prognosis. However, the SPRR3's role in suppressing EMT, as well as its mechanism(s) of action in OSCC, require more in-depth investigation.

Gene-set enrichment analysis enrichments analysis indicates that a series of metabolic processes are impacted by SPRR3. This observation is consistent with other results we acquired by $\chi^2$ test, where alcohol consumption is correlated with SPRR3 expression. Particularly, alcohol consumption is an important risk factor for OSCC, and the individual's ability to metabolize alcoholic beverages somehow reflects on its ability to control cancer. Accordingly, ethanol has been classified as a carcinogenic substance in humans (*Zygogianni et al., 2011*). Additionally, SPRR3 may be involved in the metabolism of other substances, including fatty acid, drugs, and xenobiotics. It has been reported that ADH, ALDH and CYP2E1 are indispensable in some metabolic processes (*Jin et al., 2013*). Thus. the relationship between SPRR3 and ADH, ALDH, CYP2E1 can confirm, to some extent, the role of SPPR3 in metabolic processes.

Altogether, our current results reflect the close relationship between SPRR3 and OSCC carcinogenesis and development. In addition, this work provides novel insights for follow-up studies, which should better dissect the impact of SPRR3 in OSCC, at the molecular and cellular level.

## CONCLUSION

In our research, the expression of SPRR3 was detected to be dysregulated in OSCC by means of RNA-seq, and further was verified by Bioinformatics methods. SPRR3 could be a potential bio-marker for identifying OSCC, and its under-expression could predict the prognosis of patients with OSCC. Above all, SPRR3 could be a qualified diagnostic/prognostic biomarker for OSCC.

### Funding

This study is funded by the Shandong province Tai Shan scholars project special funds, Fundamental Research Funds for the Application of stem cells and tissue regeneration in dental implant (ts201511106). It was also supported by The Youth Scientific Research Funds of School of Stomatology, Shandong University (2018QNJJ01); Shandong Medical and Health Science and Technology Development Plan (2017WS112). The funders had no role in study design, data collection and analysis, decision to publish, or preparation of the manuscript.

### Grant Disclosures

The following grant information was disclosed by the authors:
Shandong Province Tai Shan scholars project.
Fundamental Research Funds: ts201511106.
The Youth Scientific Research Funds: 2018QNJJ01.
Shandong Medical and Health Science and Technology Development Plan: 2017WS112.

### Competing Interests

The authors declare that they have no competing interests.

### Author Contributions

- Lu Yu performed the experiments, analyzed the data, prepared figures and/or tables, authored or reviewed drafts of the paper, and approved the final draft.
- Zongcheng Yang performed the experiments, analyzed the data, prepared figures and/or tables, authored or reviewed drafts of the paper, and approved the final draft.
- Yingjiao Liu analyzed the data, prepared figures and/or tables, and approved the final draft.
- Fen Liu analyzed the data, authored or reviewed drafts of the paper, and approved the final draft.
- Wenjing Shang analyzed the data, prepared figures and/or tables, and approved the final draft.

- Wei Shao analyzed the data, authored or reviewed drafts of the paper, and approved the final draft.
- Yue Wang analyzed the data, prepared figures and/or tables, and approved the final draft.
- Man Xu analyzed the data, authored or reviewed drafts of the paper, and approved the final draft.
- Ya-nan Wang conceived and designed the experiments, authored or reviewed drafts of the paper, and approved the final draft.
- Yue Fu conceived and designed the experiments, authored or reviewed drafts of the paper, and approved the final draft.
- Xin Xu conceived and designed the experiments, analyzed the data, authored or reviewed drafts of the paper, and approved the final draft.

## Human Ethics

The following information was supplied relating to ethical approvals (i.e., approving body and any reference numbers):

Research Ethics Committee of Stomatological Hospital of Shandong University approved the study (No. 20190205).

## Data Availability

Data is available at GEO: GSE140707.

## Supplemental Information

Supplemental information for this article can be found online at http://dx.doi.org/10.7717/peerj.9393#supplemental-information.

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
