# Peer review of "Identification of SPRR3 as a novel diagnostic/prognostic biomarker for oral squamous cell carcinoma via RNA sequencing and bioinformatic analyses"

_PeerJ, doi:10.7717/peerj.9393_

## Round 0.1 · original submission · Major Revisions

Please revise the paper according to the reviwers’s suggestions.

Reviewer 1 ·

Basic reporting

This manuscript provides a proof-of-concept investigation on the role of SPRR3 in OSCC by sophisticated bioinformatic analysis of three OSCC patient samples. Although a few language errors do not hinder the reader’s understanding, I do recommend the authors re-polish their manuscript or have a native speaker proof-read this manuscript before resubmission.

Experimental design

The experimental design need to be addressed more precisely.
1. Authors collected three OSCC patients sample for sequencing but why author choose mouse genome as the mapping reference (line 112)??
2. In figure 2, author should clearly label the row and column in the heatmap and point out which one is SPRR3 clearly in both volcano plot and heatmap.
3. Why author chose p value (rather than FDR) as one of the statistical cut-off?
4. Author should merge table S3 and S4 together and also list clearly logFC, FDR and p value of each DEGs.
5. GSEA should also be considered to cross-validate that findings from GO and KEGG analysis from their own 6 matched samples.
6. In line 274, how do author determine the “low expression of SPRR3”? median cut-off?
7. Authors performed Cox and LASSO regression in GSE42743 and they got SPRR3 was the most significant genes. Afterwards, they validated SPRR3 in TCGA by univariate and multivariate Cox regression analysis. This design might be confusing and mess. Why not authors perform univariate and multivariate Cox regression analysis and LASSO in one dataset, then validate what they found in anther?
8. What kind of TCGA data was downloaded ? level 3 RNA seq or microarray? What is the meaning of “the relative expression” in the y axis label of figure 6?
9. In panel I of figure 6, the label is missing.
10. In GSEA, how did author divide the patients into groups for analysis. Authors mentioned “according to the expression of SPRR3”, but more details are required.

Validity of the findings

Authors performed IHC to validate what they found in bioinformatic analysis.
1. In figure 4, each p value should be labelled in panel A. What does the star represent for in panel B and C? What kind of statistical analysis used in these heatmap?
2. In figure 6 and according text, author should be cautious about “correlated”. Authors found the expression of SPRR3 is different between the subgroup of patients, but author could not draw that was a correlation. If authors would like to test the correlation, regression (or correlation ) analysis should be considered.

Additional comments

The identification of potential molecular biomarkers in cancer is relevant to improve its diagnosis and treatment. In this line, the bioinformatic approach described in this work is useful to detect target genes with an interesting functional profiling in the frame of OSCC.

However this manuscript needs a better explanation related to three key points: 1) an adequate details about the design used that ensures the validity of findings and 2) improve the description of methods that allows the reproducibility of the work, 3) more validation works (whether the key genes from the results of GO/KEGG or GSEA are positively correlated with SPRR3).

·

Basic reporting

1. The deposition of the generated data is better mentioned right after the description of the data in the materials and methods section.
2. The introduction states that SPRR3 was “determine as our gene of interest and the objective of this study was to verify its diagnostic and prognostic value ...” based on the known function of the gene in other tumors. However, the rest of the study presents the findings as being identified by screening for more important gene.
3. No documentation of related to the use of human samples is attached to the manuscript
4. The section on differential expression analysis details the steps of the analysis (first paragraph) and then describes the data collected from different sources and how it was processed. The authors may consider writing about data collection and data processing in separate subsections.
5. The criteria of significance of differentially expressed genes doesn’t include adjustment for multiple testing. The criteria of significance in enrichment analysis is not described at all.
6. Data (GSE42743 and TCGA), annotations(mm10 genome, GO and KEGG), and tools (HISAT2, HTSeq, R, Bioconductor, gplots,, affy, imput, edgeR, DAVID, Firebrowse and Cytoscape) were not cited properly in the text.
7. Figure 2A. The volcano plot has a peculiar shape. Namely, many genes with high log2 fold change have large p-values. This might be a result of inappropriately including more than one variables in the model for the differential expression analysis. Also, the significance criteria for the fold-change and p-values are different than what is declared in the text.
8. The axis labels in Figures 2A, 2B, 4A and 6I are missing.
9. The text and the references (in line citation) would benefit greatly from another round of revision and editing. On several occasions, the meaning of the sentences are not clear because of inappropriate word use or grammatical errors.

Experimental design

1. Several datasets from GEO were used in the analysis without describing what they represent or why they are relevant to the current analysis.
2. The results of the differential expression and GO/KEGG enrichment cannot be trusted.
3. The meaning of MCODE score is not described and is not clear why it should be significant in this context.
4. Is the differential expression presented in Figure 4A independent of the one presented before? Figure 4B. presents uses two datasets from GEO without explaining what they represent or why the add value to the analysis. The same could be said about the analysis presented in Figure 5A&B.
5. The results of the survival analysis (Figure 5C&D) are different. Can the discrepancies be explained? Why isn’t the regression analysis (Table 3) applied to the GEO dataset?

Validity of the findings

The findings presented in this manuscript cannot be considered reliable enough to support the conclusions. Firstly, it is not clear whether the authors are using the workflow to arrive at the most significant gene or to justify their prior choice of SPRR1 as interesting based on previous literature. Secondly, there are several issues with the design and reporting of the study. Even when disregarding the first point, some of the analysis steps were not fully described, or missing.

Reviewer 3 ·

Basic reporting

This paper presents an equivalent method to identify the prognostic and diagnostic biomarker SPRR3 for OSCC, and builds up an experiment system for validation. Even though this paper contains interesting results which merit publication, for the benefit of the reader, however, should like to suggest that the author seek the advice of someone with a good knowledge of English, preferably native speaker to help proofread and reorganize sentences to improve the professional scientific expression.
Furthermore, a number of points need clarifying and certain statements require further justification.

Experimental design

This is a carefully done study, advanced bioinformatics tools were utilized to address specific questions, however, a number of points need clarifying and certain statements require further justification.
1. In line 44, the author claimed SPRR3 as “a potential tumour suppressor gene”, but there is no following up reference or explanation about the role that SPRR3 played as a tumor suppressor, more details are supposed to be included in the background part.
2. Line 260-261 mentioned that “SPRRs was reported to be functional”, what function??? Please provide more details.
3. Some of the figures haven’t been clearly labeled, e.g. figure 2B heat map, figure 6I
4. In 3.6 the author claimed that low expression of SPRR3 is associated with poor prognosis, however, low expression and high expression should have been defined in the first place

Validity of the findings

The work is meaningful in terms of OSCC diagnosis and potentially provides an indication for OSCC treatment. The conclusion "lower expression of SPRR3 associated with poor prognosis" has been well stated, expression in transduction level and protein level have been provided. However, the explanation in the functional aspect is still not very convincing and more evidence is supposed to be provided.

Additional comments

The work of identifying novel biomarkers for cancer diagnosis and prognosis is meaningful, this paper presents an equivalent method to identify the prognostic and diagnostic biomarker SPRR3 for OSCC, and advanced bioinformatics tools and methods have been applied. However, the main issues are as follows,

1. The scientific expression should be better polished by someone with a good knowledge of English, preferably a native speaker to help proofread.
2. More evidence demonstrates the role that SPRR3 played in tumorigenesis is supposed to provide to make the conclusion more convincing
3. Make sure all the figures are clearly labeled to help readers better understand

---

## Round 0.2 · Minor Revisions

Please see the two reviewer's comments for revisions.

Reviewer 1 ·

Basic reporting

/

Experimental design

/

Validity of the findings

/

Additional comments

The revised manuscript is much more improved and the workflow is explicitly displayed.
If authors used trim galore to remove the adapter of the raw reads, this information should be added. The key parameters of trim galore and HISAT2 are also disclosed.

·

Basic reporting

no comment

Experimental design

no comment

Validity of the findings

no comment

Additional comments

The revised manuscript addresses most of the issues raised in the review. In particular, the motivation of the study was further explained and many of the methodological details were added. Moreover, proper citations were added to the tools and datasets that were used in the study. Finally, references to the data repository where the raw data would be hosted was provided and documentation for the use of human samples were submitted to the journal.
One critical issue remains however. The initial step in the workflow, namely the differential expression analysis doesn’t inspire confidence. The shape of the volcano plot (the only provided diagnostic) is off. It’s unlikely that the small sample size is the reason. Given the importance of this step, especially because the downstream steps are all based on its results, a proper investigation should be conducted. For example, performing the differential expression with alternative models (design matrices) and showing more analysis prognostics.

---

## Round 0.3 · accepted · Accept

The manuscript is now accepted by PeerJ. Congratulations!